# Cordycepin Triphosphate as a Potential Modulator of Cellular Plasticity in Cancer via cAMP-Dependent Pathways: An In Silico Approach

**DOI:** 10.3390/ijms25115692

**Published:** 2024-05-23

**Authors:** Jose Luis Gonzalez-Llerena, Bryan Alejandro Espinosa-Rodriguez, Daniela Treviño-Almaguer, Luis Fernando Mendez-Lopez, Pilar Carranza-Rosales, Patricia Gonzalez-Barranco, Nancy Elena Guzman-Delgado, Antonio Romo-Mancillas, Isaias Balderas-Renteria

**Affiliations:** 1Laboratory of Molecular Pharmacology and Biological Models, School of Chemistry, Autonomous University of Nuevo Leon, San Nicolas de los Garza 66451, Mexico; jose.gonzalezll@uanl.edu.mx (J.L.G.-L.); bryan.espinosardr@uanl.edu.mx (B.A.E.-R.); daniela.trevinoal@uanl.edu.mx (D.T.-A.); patricia.gonzalezbrn@uanl.edu.mx (P.G.-B.); 2Center for Research on Nutrition and Public Health, School of Public Health and Nutrition, Autonomous University of Nuevo Leon, Monterrey 66460, Mexico; luis.mendezlop@uanl.edu.mx; 3Laboratory of Cell Biology, Northeast Biomedical Research Center, Mexican Social Security Institute, Monterrey 64720, Mexico; carranza60@yahoo.com.mx; 4Health Research Division, High Specialty Medical Unit, Cardiology Hospital N. 34. Mexican Social Security Institute, Monterrey 64360, Mexico; nancyegd@gmail.com; 5Computer Aided Drug Design and Synthesis Group, School of Chemistry, Autonomous University of Queretaro, Queretaro 76010, Mexico

**Keywords:** cordycepin, molecular docking, molecular dynamics, mechanism of action, purine metabolites, tumor microenvironment, adenylate cyclase

## Abstract

Cordycepin, or 3′-deoxyadenosine, is an adenosine analog with a broad spectrum of biological activity. The key structural difference between cordycepin and adenosine lies in the absence of a hydroxyl group at the 3′ position of the ribose ring. Upon administration, cordycepin can undergo an enzymatic transformation in specific tissues, forming cordycepin triphosphate. In this study, we conducted a comprehensive analysis of the structural features of cordycepin and its derivatives, contrasting them with endogenous purine-based metabolites using chemoinformatics and bioinformatics tools in addition to molecular dynamics simulations. We tested the hypothesis that cordycepin triphosphate could bind to the active site of the adenylate cyclase enzyme. The outcomes of our molecular dynamics simulations revealed scores that are comparable to, and superior to, those of adenosine triphosphate (ATP), the endogenous ligand. This interaction could reduce the production of cyclic adenosine monophosphate (cAMP) by acting as a pseudo-ATP that lacks a hydroxyl group at the 3′ position, essential to carry out nucleotide cyclization. We discuss the implications in the context of the plasticity of cancer and other cells within the tumor microenvironment, such as cancer-associated fibroblast, endothelial, and immune cells. This interaction could awaken antitumor immunity by preventing phenotypic changes in the immune cells driven by sustained cAMP signaling. The last could be an unreported molecular mechanism that helps to explain more details about cordycepin’s mechanism of action.

## 1. Introduction

Cellular plasticity can be defined as the capacity of cells to alter their phenotype and cell type dynamically based on signals received from the environment, such as hormones, nutrient availability, oxygen levels, extracellular matrix (ECM) components, and other factors [1]. Cellular plasticity has been highlighted as an essential component of the development of malignancies [2]. An example of cellular plasticity in cancer is the epithelial-mesenchymal transition (EMT), which consists of the phenotypic shift of an epithelial cell toward a mesenchymal cell, a process observed during metastasis [3]. These phenomena are related to acquiring stem cell characteristics significantly influencing cancer progression and its responsiveness to treatments [4,5]. In solid malignant tumors, along with cancer cells, various types of cells can be found, such as immune and stromal cells, such as cancer-associated fibroblasts (CAFs), and endothelial cells. Together with the ECM, these cells constitute the tumor microenvironment (TME), where cells employ messenger molecules like hormones, cytokines, and metabolites to trigger phenotypic changes in an autocrine and paracrine manner [6]. These alterations enable the cells to adapt to challenging environments and play a critical role in tumor behavior and therapeutic outcomes [7]. Adjuvant therapies like epigenetic drugs and inhibitors of specific signaling pathways have been proposed to prevent resistance or disease relapses [8].

Cyclic adenosine monophosphate (cAMP) is a crucial molecule in cellular signaling that encompasses a wide range of functions, including the phenotypic changes of cells [9,10]. This second messenger is generated upon activation of G protein-coupled receptors (GPCRs) associated with the Gα_s_ subunit, which activates the adenylate cyclase (AC) enzyme. This enzyme uses the hydroxyl group at the 3′ position of ATP’s ribose to form a cyclic structure [11]. In TME cells, we find two critical targets of cAMP: Protein Kinase A (PKA) [12] and the Exchange protein directly activated by cAMP (Epac1/2) [13]. Some molecules that can increase cAMP concentration within TME cells are adenosine [14], catecholamines [15], prostaglandin E_2_ (PGE_2_) [16], and protons (H^+^) [17], which can bind to GPCRs coupled to Gα_s_. It is noteworthy that TME is distinguished by a high concentration of these specific molecules and their corresponding receptors [18,19,20,21]. The last has been linked to the reprogramming of tumor cells toward more aggressive and immunosuppressive phenotypes promoting cancer advance [22,23,24,25]. Currently, compounds have been designed to modulate its signaling, and some have been probed in synergy with cancer immunotherapy, showing promising results in reducing tumor progression and revealing new therapeutic approaches [26,27,28,29]. Although the role of cAMP in cancer is not currently fully understood, a great deal of effort has been made to create inhibitors of both its activators and effectors [29]. (Consult Appendix A for recommended readings). 

In some cancer cells, cAMP can induce heightened proliferation, survival, stemness, invasion, and migration [30,31,32]. For instance, Rap1, an Epac effector, can activate MEK and ERK1/2 pathways [33], leading to the expression of EMT markers such as SLUG, ZEB1, TWIST1, and SNAIL1 [34]. Moreover, markers of EMT have been linked with the expression of stem cell markers Oct4, Sox2, Klf4, c-Myc, and Nanog [35,36,37] and are consistently found in patients with worse prognosis [38]. On the other hand, in immune cells, one of the principal mechanisms of immunosuppression involves the inhibition of NF-κB signaling through the cAMP-PKA-CREB axis [39,40]. For instance, the cAMP-PKA-CREB axis induces macrophage polarization from an M1 to an M2 phenotype, releasing tolerogenic and prometastatic factors such as IL-10 and TGF-β [41,42]. In the same vein, activation of the Gα_s_-cAMP-PKA axis is known to be responsible for cancer immunotherapy failure and has been proposed as a druggable target [43]. In general, cAMP signaling is crucial in the resolution of inflammation [44]. In other cells of TME, like in CAF, cAMP-mediated signaling can activate PKA and Epac, which can phosphorylate CREB and C/EBPβ, respectively [45,46]. This increases the expression of IL-6 by CAF, which is related to increased inflammation that leads to phenotypic changes in cancer cells to acquire stemness traits [47,48,49]. In endothelial cells, it can also promote angiogenesis [50]. Another protein overexpressing CAF by cAMP signaling is Wnt5a, which induces angiogenic processes and malignancy acquisition [51]. In endothelial cells, the cAMP-PKA-CREB axis leads to epigenetic changes by the activity of histone deacetylase 2 (HDAC2), which represses the expression of Thrombospondin 1 (TSP-1), a potent angiogenesis inhibitor [52]. 

Cordycepin (COR) is an adenosine (ADO) analog with a broad spectrum of biological activities, encompassing antiproliferative, immunomodulatory, and antimetastatic activities (among others), both in vitro and in vivo [53]. Clinical trials on humans are available (e.g., NCT00003005, NCT00709215). It has been reported that COR can modulate different signaling pathways, such as diminishing MEK-ERK and AKT-mTOR pathways and activating AMPK [53]. Another essential signaling pathway affected by cordycepin is the NF-κB, leading to a potent anti-inflammatory effect. A study conducted by Lee et al. (2019) revealed that COR is absorbed in the form of a previously presumed inactive product, 3′-deoxyinosine (3′-dINO), which can be converted into 3′-deoxyATP, also known as cordycepin triphosphate (COR-TP), within macrophages [54] and in tumor cells [55,56]. It is postulated that COR-TP serves as a pivotal metabolite underlying the biological effects of COR. Notably, COR and its derivatives lack the hydroxyl group at the 3′ position of the ribose, in contrast to ADO and its derivatives, maintaining the remaining structure. This enables them to interact with the same targets. Furthermore, this is well supported by experimental data; for example, we can find reports of 3′-deoxy-s-adenosyl methionine (3′-dSAM) formation [57], an analog of s-adenosylmethionine (SAM) in human cells, and other metabolites like 3′-deoxyinosine monophosphate (3′-dIMP), cordycepin monophosphate (COR-MP), and cordycepin diphosphate (COR-DP) in different biological models [55,58,59]. 

Currently, bioinformatic databases have user-friendly interfaces created to facilitate a precise structural correlation between small molecules and their potential therapeutic targets, such as SwissADME and SwissTargetPrediction [60,61]. On the other hand, molecular docking studies and molecular dynamics (MD) simulations have made it possible to validate hypotheses about sites and forms of binding obtained by molecular docking studies and estimate a theoretical affinity with greater precision [62,63,64]. For all the above, we hypothesized that COR-TP may facilitate a decrease in cAMP generation by acting as a false substrate for the AC enzyme. The absence of a 3′ hydroxyl group makes it impossible to cycle the nucleotide, suggesting a possible unreported mechanism for COR. The potential interaction could partially explain the broad spectrum of molecular effects reported by other research groups after COR administration.

To test our hypothesis, in this study, we conducted a comprehensive physicochemical analysis using chemoinformatics and bioinformatics tools by purchasing the characteristics of COR and derivatives with endogenous purine-derived metabolites. The analysis included a molecular docking study with different targets involved in purine metabolism. In addition, we performed molecular docking studies specifically between COR-TP and the soluble isoform of AC (sAC or ADCY10) to observe the potential interaction between COR-TP and the AC enzyme. Moreover, due to the limitations of molecular docking studies and the relevance of this interaction, the molecular docking study was followed by MD simulations with both one transmembrane AC isoform (tmAC), specifically ADCY05, and the soluble isoform ADCY10. This study’s significance lies in contributing to elucidating cordycepin’s complete mechanism of action by emphasizing the possible effects mediated by its derived metabolite COR-TP in cancer cells and the tumor microenvironment. Understanding cordycepin’s complete mechanism of action is crucial for its appropriate use in the clinical treatment of different pathologies. This study aims to explore how COR works and can provide a basis for future experimental studies to comprehend its mechanism of action completely, focusing on other pathways not revised before. 

## 2. Results

### 2.1. Database Creation

Before the comparison of the structures, a database was created (Table 1). COR and its derivatives, such as 3′-dINO, 3′-dIMP, COR-MP, COR-DP, COR-TP, and 3′-dSAM, were included in the database as exogenous ligands compared with endogenous molecules. As endogenous ligands, we included metabolites related to purine metabolism, such as adenosine (ADO), guanosine (GUA), inosine (INO), and their mono-, di-, and triphosphorylated forms (AMP, ADP, ATP, GMP, GDP, GTP, IMP, IDP, ITP). Also, we included the cyclic forms of adenosine monophosphate and guanosine monophosphate, cAMP, and cGMP, as well as SAM. Before the following analysis, the composite ID (CID) and the simplified molecular input line entry system (SMILES) were collected. Metabolites were considered at a physiological pH of 7.4.

### 2.2. Structural Comparison of Cordycepin and Its Derivatives against Purine Derivatives

We used the PubChem Score Matrix server to evaluate the structural similarity, both in 2D and 3D, between COR and its derivatives relative to endogenous nucleosides and nucleotides. The analysis results are presented in Table 2. Initially, 2D comparisons were conducted to determine the Tanimoto coefficient (TC), which measures the degree of similarity between 2D representations. These parameters yield values ranging from 0 to 100, with 100 indicating maximum 2D similarity. As expected, the comparison between COR and ADO yielded a TC value of 98; the comparisons of COR-MP/AMP, COR-DP/ADP, and COR-TP/ATP yielded a TC of 99. Likewise, the comparison between INO/3′-dINO generated a TC value of 99. In addition, we observed a decrease in the TC compared to 3′-dINO/ADO (TC of 77) and INO/ADO (TC of 78). 

Regarding the 3D comparison, we analyzed similarity using Color Tanimoto (CT) and Shape Tanimoto (ST) parameters. These parameters yield values ranging from 0 to 100, with 100 indicating maximum 3D similarity. The ST value assesses the shape and orientation that molecules can adopt in space. Simultaneously, the CT evaluates distinctive chemical features such as aromatic rings, hydrogen bond donors or acceptors, and similar functional groups. After obtaining the ST and CT values, we added them together to calculate the Combo T parameter. The Combo T parameter ranges from 0 to 200. According to the literature, ST, CT, and Combo T values above 80, 50, and 130 suggest acceptable similarities [65]. COR and its derivatives notably maintained higher scores regarding 3D similarity than endogenous metabolites. Specifically, the COR/ADO comparison yielded an ST value of 98, a CT of 85, and a Combo T of 183. For COR-MP/AMP, we observed an ST of 91, a CT of 65, and a Combo T of 156. Likewise, COR-DP/ADP exhibited an ST of 77, a CT of 55, and a Combo T of 132, while COR-TP/ATP showed an ST of 86, a CT of 43, and a Combo T of 129. The various comparisons between COR and ADO, whether in nucleoside or nucleotide form, demonstrate robust 3D structural similarity in shape and functional characteristics. In the INO/3′-dINO comparison, ST values were 94, CT was 63, and Combo T was 157. The 3′-dINO/ADO comparison yielded an ST value of 98, a CT of 52, and a Combo T of 150, surprisingly higher even than the compared values of inosine/adenosine (ST of 89, CT of 34, and a Combo T of 123).

### 2.3. Physicochemical Comparison of Cordycepin and Its Derivatives against Purine Derivatives

According to the hierarchical clustering (HC) between COR-TP (Figure 1) and the metabolites included in our database, eight clusters were generated. Cluster 1 showed a percentage distribution of 9.21%, including GDP, ADP, IDP, ATP, GTP, ITP, and COR-TP. All these metabolites were represented by two features: they were purine nucleotides with two or three phosphate groups. On the other hand, cluster 2, showed a percental distribution of 17.11%, and it included IMP, AMP, GMP, cAMP, and cGMP, all being monophosphorylated purine nucleotides. However, cluster 2 also included purine nucleosides (ADO, INO, and GUA), glycolytic intermediates (GLU, G6P, F6P, and F1,6BP), and L-ASS. Cluster 3 with a percentage distribution of 1.32% was composed of PAL, the only fatty acid in our database that differed markedly from the rest of the metabolites. Cluster 4 showed a percentage distribution of 10.53% and was composed of intermediates from glycolysis (DHAP, G3P, 1,3BPG, 3PG), TCA cycle (MAL, ISOCITR, CITR), and GLI3P. Cluster 5 showed a percentage distribution of 18.42% and was composed of metabolites of creatine metabolism and urea cycle (L-CAR, L-ORN, L-LYS, PUT, CAD, AGM, SPD, SPR, ADMA, L-CIT, SDMA, L-ARG, NG-MMA, L-HARG). Cluster 6 showed a percentage distribution of 17.10% and was miscellaneous in composition, including compounds from the metabolism of amino acids, glucose, creatine, and polyamines in addition to some intermediary metabolites of urea and TCA cycles (L-TYR, L-KYN, L-TRP, L-PHE, FUM, CP, PEP, PYR, ACETO, OXA, SUCC, KETOG). Cluster 7 showed a percentage distribution of 10.53% and was composed mainly of compounds from the metabolism of amino acids, glycolytic intermediates, and ketone bodies (L-VAL, L-PRO, L-MET, L-LEU, L-ILE, ACET, LAC, 3OHB). Finally, cluster 8 showed a percentage distribution of 15.79% and was composed of compounds from the metabolism of amino acids and creatine (L-SER, L-THR, L-GLY, L-CYS, L-ALA, CREA, GAC, L-HIS, L-ASP, GLUT, L-ASN). 

Furthermore, clustering was also performed on the other COR derivatives, resulting in the same eight clusters with minor variations. The only changes observed were in the cluster of interest, which varied according to the unique characteristics of each molecule. For instance, COR-DP clustered with ADP, GDP, and IDP; COR-MP was grouped with AMP, GMP, IMP, cAMP, and cGMP. COR was grouped with ADO, GUA, and INO; 3′-dINO also clustered with GUA, ADO, and INO (Appendix A). All metabolites were correctly grouped based on their phosphate numbers. It is important to note that 3′-dINO differs only in the nitrogenous base from COR, hence their grouping with the same element set. Additionally, in the specific case of COR-MP, it is more remarkable. The rest of the percentage distributions for each clustering can be found in the Appendix A. In accordance with the findings of this analysis, it is evident that COR derivatives, as dictated by their physicochemical properties, may exhibit a relationship with endogenous metabolites.

### 2.4. Target Prediction According to Structural Characteristics of Cordycepin and Its Derivatives

To test whether the observed structural and physicochemical similarities between COR and its derivatives with the relevant endogenous metabolites translated into an affinity for similar targets, we conducted a target prediction based on the structure using the Swiss Target Prediction and Similarity Ensemble Approach (SEA) tool (Appendix A). The results generally indicated that COR derivatives bind to targets associated with the purinergic and adenosinergic system, enzymes involved in nucleotide metabolism and salvage pathways, specific kinases, and enzymes like DNA polymerase and sAC. These findings align with the 2D, 3D, and physicochemical similarities obtained earlier, as all the mentioned targets interact with endogenous ligands such as ADO, AMP, ADP, and ATP or molecules related to nucleotides like SAM.

### 2.5. Comparative Analysis of Affinity and Binding Modes of Cordycepin and Its Derivatives with Endogenous Targets

To test whether our previous results regarding the physicochemical and target prediction characteristics resulted in similar affinity between cordycepin derivatives and ADO derivatives, we performed a molecular docking study. The docking scores of COR and its derivatives and endogenous metabolites with their respective targets are shown in Figure 2. Our molecular docking results generally demonstrated comparable affinities between COR and ADO derivatives across various targets. For instance, in the adenosine kinase (ADK) enzyme site 1, we obtained values of −6.24 kcal/mol, −6.92 kcal/mol, and −5.58 kcal/mol for ADO, COR, and 3′-dINO, respectively, and values of −4.25 kcal/mol, −5.56 kcal/mol, and −5.23 kcal/mol for ADO, COR, and 3′-dINO for site 2, indicating the potential occupation of their active sites by COR and 3′-dINO. Similarly, the adenosine receptors evaluated (A_1_R, A_2A_R, A_2B_R) exhibited the same pattern, suggesting once again the possible binding of these receptors by COR and 3′-dINO. Furthermore, comparable affinity values were observed between ADO and COR in the enzyme adenosine deaminase (ADA), with docking scores of −7.84 kcal/mol and −7.94 kcal/mol, respectively. 

Regarding enzymes that utilize ATP as a substrate, we investigated ectonucleoside triphosphate diphosphohydrolase-1, also known as CD39, ecto-nucleotide pyrophosphatase/phosphodiesterase 3 (ENPP3), methionine adenosyl transferase (MAT), and ADCY10 (sAC). The latter was chosen due to its catalytic mechanism of action. We compared the affinities of ATP (the original substrate) and COR-TP with these targets. In the case of CD39, we include the comparison between COR-DP and ADP, as ADP can be a substrate for this enzyme. Regarding CD39, ATP has a value of −6.12 kcal/mol, and COR-TP has a value of −7.43 kcal/mol. ADP obtained a value of −7.51 kcal/mol and COR-DP a value of −7.92. In the case of ENPP3, ATP obtained a value of −4.15 kcal/mol and COR-TP of −5.08 kcal/mol. MAT exhibited −3.52 kcal/mol values for ATP and −2.93 kcal/mol for COR-TP. In the case of sAC (ADCY10), we observed values of −5.78 kcal/mol for ATP and −5.73 kcal/mol for COR-TP, indicating a potentially favorable interaction of the enzyme with this metabolite. The results from molecular docking studies hint that COR and its derivatives could serve as mimetic molecules of ADO and its derivatives at a biological level.

After analyzing the docking scores, the binding modes between COR and some of its derivatives, such as 3′-dINO and COR-TP with some ADO and ATP targets, were analyzed to evaluate whether the binding modes of COR and its derivatives were comparable to those of endogenous metabolites. A subset of the targets for the non-phosphorylated molecules are depicted in Figure 3, while Figure 4 presents the targets for the phosphorylated molecules. The remaining interactions can be observed in Appendix A. Regarding non-phosphorylated molecules, the binding modes for A2AR and A2BR adenosine receptors were evaluated. In addition, the ADK (site 1) and ADA were included in this analysis. We could observe comparable binding modes between COR, 3′-dINO, and the endogenous metabolite ADO in all cases. In the case of A2AR, hydrogen bonds can be observed between ADO and amino acids of the active site as Glu 169, Asn 253, and His 278; COR shows hydrogen bonds with Glu 169 and Asn 253; 3′-dINO shows hydrogen bonds with Asn 253 and Glu 169. The binding site aligns with the reported by UniProt, specifically with the amino acid residues Glu 169, Asn 253, Ser 277, and His 278. For the A2BR receptor, the endogenous ADO ligand has two hydrogen bonds with Asn 254; COR shows hydrogen bonds with Thr 89, Asn 186, and Asn 254; 3′-dINO features hydrogen bonds with Asn 254. The UniProt reported amino acid residues were Glu 174, Asn 254, Ser 279, and His 280.

For the ADA enzyme, ADO, its endogenous ligand shows interactions through hydrogen bonds with Asp 19, Gly 184, Glu 217, His 238, and Asp 295; COR forms hydrogen bonds with His 17, Asp 19, Gly 184, Glu 217, His 238, and Asp 295; 3′-dINO shows hydrogen bonds with Cys 153, Gly 184, Glu 217, and His 238. Our docking study is comparable with the amino acid residues reported by UniProt in the binding site, which were His 17, Asp 19, Gly 184, Glu 217, and Asp 295. Finally, in Figure 3, hydrogen bonding interactions with the ADK (site 1) can be observed, where the endogenous ligand ADO shows interactions with Asn 14, Asp 18, Gly 64, Ser 65, and Asp 300; COR shows interactions with Asn 14, Asp 18, Gly 64, and Ser 65; the 3′-dINO shows a similar interaction to ADO and COR but without the hydrogen bond with Asn 14. Despite the observed similarity among these ligands within the binding site, it presents a divergence from the findings reported by UniProt (Asp 35, Gln 306). This difference may be due to the limitation of our docking study, that is, the lack of magnesium ions or conformational restraints in the crystallography.

On the other hand, Figure 4 shows a comparison of the ATP and COR-TP binding modes on the targets CD39, ENPP3, and sAC and ADP and COR-DP binding modes on CD39. In the CD39 protein, we observe the formation of hydrogen bonds with the amino acids Arg 93, Arg 113, Lys 143, and Thr 164 for both the ATP molecule and COR-TP. Additionally, the latter exhibits interactions with Lys 161, Thr 237, Asp 242, and Glu 161. They obtained a total of twelve interactions for COR-TP and eight for ATP. Also, for ADP and COR-DP, we observe the formation of hydrogen bonds with the amino acids Arg 93, Arg 113, Lys 143, and Lys 161, and additionally, COR-DP presents interactions with Tyr 237 and Thr 237. The binding modes of the ligands in CD39 protein correspond with the findings reported in UniProt, where the interactions are observed at Lys 143, Thr 164, Thr 237, Asp 242, and Lys 324. Our simulation of ENPP3 with ATP and COR-TP established hydrogen bonds with Asn 227, Asn 226, and Gln 244 but differed in the hydrogen bond formation of ATP with Lys 204 and of COR-TP with Tyr 289 and His 329. The reported amino acid residues that interact according to UniProt were Lys 204, Thr 205, Asn 226, Glu 275, and Tyr 289. They obtained eight interactions for COR-TP and seven for ATP, with some differences between the residue’s interactions. Notably, the orientation and interaction between ATP and COR-TP were comparable for CD39 and ENPP3. For sAC, the binding modes between ATP and COR-TP are similar, maintaining the interaction with Asp 99 and Arg 416 and COR-TP differing with ATP in the interaction with Thr 52. According to the report by UniProt, our simulation conserved certain interactions, such as Thr 52, Asp 99, and Arg 416. As previously mentioned the lack of magnesium ions during the simulation, the orientation and binding modes of the ligands deviated from their reported positions. Consequently, several interactions, including those involving Ser 49, Lys 144, and Val 406, were significantly impacted.

However, given that molecular docking has certain limitations, such as the simulations being conducted in a rigid mode where the protein lacks movement and the absence of metal cofactors and water molecules at the binding site that could be relevant for interaction, we decided to validate the molecular docking results using a more sophisticated technique. Therefore, considering ligands ATP and COR-TP, we performed MD simulations on the soluble and transmembrane isoforms of the AC enzyme.

### 2.6. Molecular Dynamics

#### 2.6.1. Construction of Models by Homology

Table 3 shows the quality analyses performed by Molprobity [66] to verify the quality of the homology models obtained.

By having more than 90% of the amino acids in permitted rotations according to the Ramachandran plot [67], which tells us the rotations of the dihedral angles of the amino acids of the known proteins, we can consider that the models obtained by the RobettaFold algorithm meet sufficient quality criteria to be able to continue being used in the calculations. Additionally, the Z-score tells us that the conformation of the models obtained is less than two standard deviations from the conformations obtained by crystallography, which confirms the model’s good quality. The systems were directly constructed for relaxation using these models by MD simulations, which are presented in the next section.

#### 2.6.2. Relaxation of Complexes by Molecular Dynamics Simulations

Once the nucleotide–enzyme complexes were constructed, they were relaxed for 100 ns in all-atom simulations. These simulations were run to stabilize the protein–ligand complexes and thus have a more representative conformation (C1) of this simulation. Figure 5 shows the change in RMSD concerning time for both complexes and ligands.

Despite apparent fluctuations, systems with ADCY05 stabilize (little RMSD change) in the presence of ligands from 50 ns. This is not the case with ADCY10, where the free system has a higher RMSD fluctuation than the complexes, and the lowest conformational fluctuation occurs in the presence of COR-TP. In all cases, 100 ns simulations can be considered sufficient to stabilize the systems.

In addition, a change in the position of the ligands was observed concerning the protein and the initial position of the ligand. This information is presented in Figure 6 and Figure 7.

As can be seen, ATP changes its conformation significantly in both proteins, suggesting that it does not undergo radical conformational changes from its original structure despite being bound to the binding site. In both cases, COR-TP does not undergo significant conformational changes. These subtle changes suggest that the docking pose used to build the system for MD simulations is stable.

Additionally, the trajectory of the MD simulations was clustered to extract the most representative conformation (C1) based on protein Cα RMSD. In Table 4 are the results of this clustering.

As can be seen, all trajectories yielded almost the same clustering histogram profile, and each C1 timeframe belonged to the final stages of the trajectories, where the complexes were considered stabilized. Noteworthy, the ADCY10-ATP complex C1 timeframe is significantly lower than the others; despite the seemingly drastic RMSD changes observed in Figure 5, this timeframe suggests that the trajectory could be considered stable earlier than the other complexes. On the other hand, the size of ADCY10 and topological state could be important factors in this RMSD behavior, compared to ADCY5, since the latter is smaller and embedded in a membrane, thus limiting the movement of the protein and lowering RMSD.

Considering the interactions formed and lost during the interaction, Figure 8 and Figure 9 present the amino acids that interact most frequently with the ligands. As can be seen, both ligands form predominantly ionic interactions with ADCY05, with ATP being the one that depends more on interactions mediated by water molecules to interact, which is logical given the lower polarity of COR-TP. The latter has more hydrophobic interactions, mainly with the amino acids Phe 1074 and Ile 1125, as well as hydrogen bonds with catalytic site residues. Despite the structural similarity, a different pattern of interactions is noted, both in amino acids and the type of interactions. This may suggest that both compounds will have different affinities for the catalytic site of ADCY05.

In the case of ADCY10, there is also a more significant number of hydrophobic interactions in the complex with COR-TP. However, there are not as many interactions as in the other isoform. Additionally, the interactions that predominate in both cases are hydrogen bonds and interactions mediated by water molecules. It is also observed that there is a differentiated pattern in the interactions that could suggest a differential affinity of these compounds for this isoform.

#### 2.6.3. Well-Tempered Metadynamics Simulations

Metadynamics calculations were performed with the most representative conformation of the MD simulations to obtain a free energy profile based on the distance between the ligand centers of mass and the amino acid binding site. Figure 10 shows the energy profiles of the ligand–cyclase complexes.

To obtain this free energy profile, the distance between these centers of mass (collective variable) is changed. By applying Gaussian potentials, conformational changes that would otherwise be difficult to obtain due to the high computational cost are achieved. Therefore, this distance-based free energy profile can help us find the binding free energy between two molecules, as presented in Table 5.

As shown in Figure 10, the free binding energy profiles are very similar between the molecules, which could be expected given the high structural similarity of the compounds evaluated on the same protein. However, as observed from the analysis of intermolecular interactions, the subtle structural differences were sufficient to give differential affinity between ligands. As shown in Table 5, the ligands showed more binding affinity (more negative energy) to ADCY05 than to ADCY10, offering the same pattern: COR-TP is more akin to the adenylate cyclase studied than the natural substrate. Although the same trend is observed in both cyclases, it is worth mentioning that for ADCY05, COR-TP is more than 50 times more akin to ADCY05 than the natural substrate, ATP. The affinity difference is less than seven times in ADCY10, and even the difference in free energy does not exceed the classical uncertainty of computational methods (1 kcal/mol), so it could even be suggested that there is no significant difference between the ligands in ADCY10.

## 3. Discussion

It has been demonstrated that COR-TP can be formed in vitro in tumor cells [55,56] and immune cells such as macrophages [54]. There is evidence that COR (not COR-TP) can weakly inhibit the enzyme adenylate cyclase [68], although the mechanism has yet to be elucidated fully. Our study confirmed that this inhibition may occur due to decreased AC activity mediated by COR-TP. This mechanism is similar to the mechanisms used by RNA polymerases [56,69,70], poly(A) polymerases (PAP) [71,72,73,74], and DNA primase [75], all of which function through similar mechanisms, using the 3′-hydroxyl group as a nucleophile. In addition to the available experimental evidence, our findings were supported by the results obtained from physicochemical comparisons between COR and its derivatives against ADO and its derivatives. Similarly, we conducted molecular docking studies of COR and its derivatives with targets interacting with ADO and its derivatives. We found that they have similar binding modes and comparable or even improved binding energies in some instances (Figure 2 and Figure 3). Finally, we performed MD simulations using two isoforms of AC (tmAC and sAC), comparing the interactions of COR-TP/AC with those of ATP/AC. 

Within the fundamental aspects to consider for drug discovery and development, the candidate molecule must possess specific physicochemical characteristics. These include having certain hydrogen bond acceptor or donor groups, a particular shape and size, flexibility, planarity, polarity, and charges in a manner that allows optimal interaction with the target [76]. The methodology employed in this study has been utilized to elucidate novel or unexplored mechanisms of action, as documented by Espinosa et al. (2023) [77]. Our results from physicochemical comparisons support that COR and its derivatives may be recognized by targets that also recognize ADO and its derivatives due to their significant similarity. Interestingly, these targets cannot distinguish between endogenous ligands, COR, and its derivatives. This conclusion is further supported by experimental evidence, where compounds such as 3′-dINO [54], COR-MP, COR-DP, COR-TP [55,56], 3′-dSAM [57], and 3′-dIMP [58,59] are formed in biological models. Notably, the presence of 3′-dIMP suggests that 3′-dINO could be recognized and phosphorylated by enzymes such as adenosine kinase, as indicated by our physicochemical and molecular docking studies. 

Regarding in silico works, a molecular docking between COR-MP and AMPK was published by Wang et al. (2010) [78], in which comparable binding energies and union modes were reported concerning the interaction of AMP and AMPK. Later, this was verified in vitro by Hawley et al. (2020) [56], where it was found that the activation of AMPK by COR was mediated mainly by its monophosphate metabolite (COR-MP). Regarding this, our results were similar to those reported by Wang et al. (2010) where, as expected, we found comparable binding affinities and union modes; however, in our study, some targets showed higher affinity for COR or its derivatives. On the other hand, Niramitranon et al. (2020) [79] also conducted MD simulations between COR and an ADA isoform. They reported that COR displayed a comparable affinity, although lower than ADO, the endogenous ligand. In our study, one isoform of AC obtained a comparable affinity (ADCY10), and the other obtained a higher affinity (ADCY05) when comparing endogenous against exogenous ligands. To date, limited in silico studies have explored the interactions between COR derivatives and human targets, particularly in comparison to endogenous metabolites, an area worthy of further exploration. 

COR-TP formation is not forced on all cells, possibly only those with accelerated metabolism and rapid proliferation, such as cancer, immune [80], and stem cells [81], which share a similar metabolism [82]. This would explain the low toxicity of COR and the differences in its mechanism of action since not all cells are forming the metabolite triphosphate; therefore, its effect is tissue-specific [54]. It is noteworthy that in the tumor, there is a high metabolic and signaling activity by the adenosinergic system [83], and remembering the structural similarity between this and COR, it is not difficult to suppose that some enzymes recognize it as ADO, fortuitously generating COR-TP in a site where it is most needed without causing severe side effects. Our MD simulations demonstrate that COR-TP can bind effectively to tmAC (ADCY05) and sAC (ADCY10). 

Although there exist several reports on the mechanism of action of COR, including its modulation of various signaling pathways, such as the reduction in EGFR phosphorylation and downstream targets (PI3K-AKT), AMP-activated protein kinase (AMPK) activation, mechanistic target of rapamycin (mTOR) inhibition, induction of death receptor-mediated apoptosis, and decreased GSK-3β phosphorylation [53,84], the complete molecular action of COR remains incompletely understood. These known mechanisms do not fully account for the extensive biological effects of COR. Notably, limited research has explored the molecular mechanisms of the dual impact of COR on the immune system: it exhibits a potent anti-inflammatory effect while simultaneously enhancing antitumor immunity in a pro-inflammatory manner. The anti-inflammatory mechanism is mainly explained by AMPK-mediated inhibition of NF-κB [85]. However, this mechanism could not explain the pro-inflammatory effects on TME since inactivation of this transcription factor would lead to an anti-inflammatory effect. We postulate that this effect is achieved by diminishing cAMP formation and concurrently enhancing the antitumor proinflammatory response through COR-TP. In addition, decreased cAMP levels in cancer cells and CAF can diminish the release of tolerogenic factors, thereby assisting the immune system in eliminating these cells [86,87]. 

The apparent contradiction between pro-inflammatory and anti-inflammatory mechanisms could be explained by the fact that COR-TP would be formed to a greater extent in the tumor microenvironment, whereas in the rest of the body, it would remain in its non-phosphorylated and deaminated form (3′-dINO), having anti-inflammatory effects by the exact immunosuppressive mechanisms of the deaminated form of ADO (INO), since it has been proven that INO can weakly activate adenosine A_2A_ receptors [88], and presenting anti-inflammatory effects [89]. The latter may partially explain, in a mechanistic way, the dual effect of COR showing anti-inflammatory and pro-inflammatory effects at the appropriate sites, on the one hand, decreasing inflammation at the systemic level and, on the other, inhibiting the anti-inflammatory phenotype in the tumor microenvironment, increasing the destruction of tumor cells by the immune system in addition to the mechanisms already reported such as the induction of apoptosis, among others. This makes COR an optimal candidate to be administered in conjunction with immunotherapy, as has already been documented and is discussed below.

If cAMP-mediated signaling were interrupted, we should observe a phenotype regression in proliferative and malignant cancer cells and a diminishing in immunosuppressive mechanisms within TME. According to our hypothesis and results, COR-TP could bind to AC and cause a decrease in cAMP concentration within the cells of the TME. Regarding the use of COR as a modulator of antitumor immunity, Deng et al. (2022) demonstrated in a murine model of colon cancer that COR can increase the infiltration of CD4^+^ and CD8^+^ T lymphocytes, NK cells, and a higher proportion of macrophages with an M1 phenotype. Moreover, it reduced the expression of CD47, promoting increased phagocytosis of cancer cells and more significant T-cell infiltration by inhibiting their apoptosis [90]. On the other hand, Chaicharoenaudomrung et al. (2023) tested in an NK cell line (NK-92) that exposure to COR increased the mRNA expression of genes encoding proinflammatory cytokines such as IL-2, TNF-α, and IFNγ and enhanced their antitumor activity against two cancer cell lines by increasing interferon and granzyme B secretion [91]. This was accompanied by a decrease in the cell surface marker CD27 (protumoral) and an increase in the expression of CD16, NKG2D, and CD11b markers that favor antitumor immune activity [91]. Following the potentiating effect of COR on immune system cells, combination therapies of COR and ICI have been conducted to improve the destruction of cancer cells. For example, Feng et al. (2023) evaluated the combined effect of COR with a CD47 inhibitor, observing that the combined effect increased the proportion of macrophages with an M1 phenotype while increasing the number of CD8^+^ T lymphocytes infiltrating the tumor [92]. 

Recently, Chen et al. (2024) published an analysis of single-cell RNA sequencing on different tumor microenvironment cells in vitro and in vivo models [93]. They observed that after the administration of COR in conjunction with a TIGIT inhibitor (an ICI), there was an increase in the number of dendritic cells infiltrating the tumor and showing an increase in the presentation of tumor antigens to CD8^+^ and CD4^+^ T cells while decreasing their interaction with Tregs promoting antitumor immunity. Another significant finding of this group was that cells treated with COR expressed the marker CD226, which is associated with higher 3-year survival of patients who lend its expression compared to patients who do not express it since its expression is necessary for antitumor immunity [94]. In addition, the combination therapy improved NK cells’ proliferation, infiltration, and toxicity in the tumor [93]. Similarly, Chen et al. (2023) demonstrated that COR combined with a CTLA-4 inhibitor had a synergistic effect by increasing the number of infiltrating effectors CD8^+^ T lymphocytes and decreasing tolerogenic FOXP3^+^ [95]. Finally, in another recently published study, Chen et al. (2024) proved that COR could be used as an adjuvant to increase the immune response after vaccination against the rabies virus in a mouse model. They observed phenotypic maturation of dendritic cells and humoral immunity enhanced with an increased number of antibodies against the virus at the serum level, reinforcing the evidence for the immunomodulatory capacity of COR [96]. Therefore, it can be observed that COR can reverse immunosuppressive phenotypes individually or in synergy with ICIs; however, to date, no mechanism relates these changes to the administration of COR.

Regarding the other cells in TME, COR has demonstrated a diminishing of the MAPK pathway primarily through decreased EGFR activation and signaling [97]. Although the precise mechanism by which COR inhibits receptor phosphorylation remains incomplete, it has been proposed that COR-TP binds to the receptor instead of ATP, avoiding phosphorylation [98]. However, based on the receptor’s mechanism, it is more probable that the COR-TP acts as an ATP molecule, donating its phosphate group. This is because the hydroxyl group at position 3′ is not essential. To date, there is no evidence of inhibition of EGFR by COR-TP based on its structure, although the possibility is not excluded. It is noteworthy that cordycepin’s capacity to inhibit EGFR signaling was comparatively less potent than positive controls like gefitinib [98,99]. The MAPK pathway can be alternatively activated via Epac and its effector Rap1, which interacts with proteins such as Raf1, Ral-GDS, and PI3K and competes with RAS for B-Raf. This continues the signaling involving MEK and ERK1/2 [100,101]. Rap1-mediated signaling influences cell survival, migration, proliferation, differentiation, and cell adhesion [102,103]. An essential way in which Rap1 promotes metastasis is through the activation of integrins, well-known players in cell migration [104,105]. 

Rap1GAP, a negative regulator of Rap1, is downregulated in various cancer types, leading to increased aggressiveness and acquisition of EMT markers [106]. Furthermore, increased Rap1GAP expression reduces EMT progression, invasion, and migration in various cancer types [106]. Rap1 mediates the downregulation of E-cadherin expression and upregulation of matrix metalloproteinases (MMPs) like MMP2 and MMP9, favoring TGF-β activation and promoting cellular migration [107]. An upregulation of matrix metalloproteinase (MMP) expression enhances processes like EMT, thereby augmenting the invasive potential of tumor cells [108]. Furthermore, it has been observed that Rap1 may stimulate Src, subsequently initiating the MAPK/ERK pathway, leading to the expression of VEGF in tumor cells and facilitating processes like angiogenesis [109]. 

Additionally, the last correlates with the expression of the immune checkpoint PD-L1 that contributes to immune system evasion [110,111]. In malignant neoplasia, these alterations induce plasticity and lead to stem-like states in cancer cells [5]. Regarding this, COR has demonstrated reduced activity in leukemia stem cells [112] and decreased stem cell traits in ovarian cancer stem cells [113]. In addition, different reports suggest that COR inhibits ERK phosphorylation mediated by MEK [98,99], but the complete mechanism is not explained. Furthermore, there are reports where COR induces an increase in E-cadherin expression and a decrease in MMP9 expression, resulting in reduced TGF-β activation and thus inhibiting the EMT process [114,115,116]. A recent study also revealed that COR decreases PD-L1 expression in a murine model of colon cancer [117]. Considering these findings, some Epac inhibitors (e.g., ESI-09) have been proposed [118]. We propose that COR reduces Epac activation by decreasing cAMP concentration, thereby preventing Rap1 activation and subsequent MAPK pathway inhibition. This effect would occur upstream in the signaling pathway, specifically via COR-TP, starting from cAMP generation. Consequently, this prevention of Rap1 activation by Epac aligns with the observed biological effects in experimental results following COR treatment. Figure 11 summarizes the mechanisms discussed so far.

On the other hand, it is essential to mention the difficulty of COR generating resistance in tumor cells due to the intrinsic molecular promiscuity and multiple reported anticancer mechanisms. The physicochemical similarity of COR and its derivatives with ADO and its derivatives, which are central molecules in energetic metabolism, cell signaling, and genetic and epigenetic processes, translates into the difficulty for cancer cells in generating resistance mechanisms. As it can be seen, reports show that COR sensitizes cells to undergo apoptosis and remodeling of the entire TME, promoting antitumor immunity. It should be noted that one of the limitations of this study is that it is an in silico approach that does not perfectly replicate what happens in biological systems. Other specific experimental evidence is needed on different types of cells and proposed targets. Another remarkable limitation was the absence of magnesium ions in some molecular docking studies, especially in enzymes that work with di- and triphosphorylated nucleotides, which may influence the predicted affinities and binding modes. However, our focus was not on determining the exact binding mode of these ligands but rather on supporting our hypothesis which held that the high structural and physicochemical similarities between COR (and its derivatives) and the endogenous metabolites derived from purine, could be extrapolated to comparable affinities and binding modes in their respective targets. Finally, it is noteworthy that docking-based methodologies have been reported to show a wide variation in their results when compared with experimental evidence [119]. For this reason, we conducted molecular docking studies with two additional tools different from the one used here, and the data can be found in Appendix A. Nevertheless, despite these limitations, our results provide the basis for new mechanisms and their plausibility. It opens a focus of study that has not been explored before to clarify the complete mechanism of action of COR. Also, it is essential to mention that not all types of cancer respond positively to the decrease in cAMP, and some even decrease proliferation when the amount of cAMP increases [120,121], so it remains a complex issue where more research is needed. However, the proposed mechanism would not be the only one by which COR acts and continues to be a promising drug candidate to continue in clinical research for cancer and other pathologies.

## 4. Materials and Methods

### 4.1. Creation of Databases

First, we selected candidate endogenous molecules related to purine metabolism due to the similarity between COR and the structural core of purines, including ADO, GUA, INO, and their phosphorylated derivatives (AMP, ADP, ATP, GMP, GDP, GTP, IMP, IDP, ITP cAMP, and cGMP), as well as SAM. The COR and its derived metabolites (COR-MP, COR-DP, COR-TP, 3′-dINO, and 3′-dIMP, 3′-dSAM) were included to complete the selection. The CID and SMILES codes were collected from PubChem for each selected molecule. In addition, we built the molecules with a physiological pH of 7.4 employing Avogadro [122]. COR and its derivatives that have been reported in the literature are shown in Table 1 in comparison with endogenous metabolites derived from ADO.

### 4.2. Structural Comparison of Cordycepin and Its Derivatives against Purine Derivatives

Once the database was created, 2D and 3D similarity analyses were performed. In the case of 3D similarity, shape and feature analysis were carried out. CIDs were introduced into the PubChem Score Matrix service [123]. In 2D similarity, the Tanimoto coefficients (TCs) were obtained that compare the fingerprints between the candidate molecule and have a range from 0 to 100, where the value 0 is related to the minor similarity and the value 100 to the highest similarity, with the highest score being the identical molecule. For 3D similarity, two scores were obtained: the Tanimoto Shape (ST) and the Tanimoto Color (CT). The ST score quantifies the steric form of the molecule versus another. In contrast, CT quantifies the chemical characteristics of the atoms in each molecule, such as hydrogen bond donors and acceptors, rings, and hydrophobic regions. Like CT, ST, and CT can have values from 0 to 100, where 0 is the null similarity. At the same time, 100 is the highest similarity related to steric space for ST and chemical characteristics for CT. However, the most frequently used score to assess 3D similarity is the Combo Tanimoto (Combo T), obtained by the sum of ST and CT. The Combo T is between 0 and 200, and, like the other parameters, 0 is the null ratio, and the value 200 corresponds to the identical molecule.

According to the literature and present work, these values can be categorized as low, moderate, and high similarity. The high category in 2D similarity presents values from 50 to 80, according to Kim, S. [65], and we consider 80 to be the cut-off point for CT. Values below 80 but above 50 were classified as moderate and low if the TC ≤ 50. For 3D similarity, ST 80 ≥ and CT ≥ 50 or Combo T ≥ 130 were considered high. The PubChem score matrix tool provided the parameters, TC, ST, and CT, of each comparison between COR and its derivatives against the metabolites of the database created. Subsequently, the Combo T was calculated.

### 4.3. Physicochemical Comparison of Cordycepin and Its Derivatives against Purine Derivatives

For analysis, we compared the physicochemical parameters of each candidate molecule predicted using the Swiss Institute of Bioinformatics SwissADME tool (http://www.swissadme.ch/ (accessed on 15 April 2023)) [60,124,125,126,127,128,129,130]. The analysis included different parameters that considered various characteristics of the molecules, such as molecular weight, sp^3^ carbon fraction, the number of rotational bonds, number of hydrogen acceptors and donors, the log P, and the TPSA, which considers molecular size, planarity, flexibility, the capacity of hydrogen bond formation, lipophilicity, and polarity, respectively. To perform the physicochemical comparison, we performed a hierarchical grouping in Rstudio using the SMILES collected from the molecules in the database that are considered to have a physiological pH of 7.4. To verify that the clusters between molecules were carried out correctly, we included a set of endogenous cell metabolites that do not belong to purine metabolism to avoid false positive clusters due to the small number of our original database. Due to the difference between the scale in the values obtained from the different parameters, the scales of magnitudes were standardized, generating a Z-score. Finally, the data were fed into a hierarchical clustering (HC) analysis for unsupervised machine learning to test whether the algorithm could recognize the relationship between candidate molecules according to physicochemical characteristics. For HC, COR derivatives were compared one by one against all metabolites in the database. The HC was performed using the R software (4.2.2 version), and the RStudio interface using the following packages: tidyverse, factoextra, cluster, ggplot2, ggcorrplot, and readr [124,125,126,127,128,129,130]. In addition, the Euclidean distances and Ward’s method were used to generate the HC clusters. It is worth mentioning that the HC analysis included different cellular metabolites with structures distinct from COR and its derivatives. These encompassed compounds from the metabolism of amino acids, fatty acids, glucose, creatine, and polyamines in addition to some intermediary metabolites of urea and TCA cycles. This approach was taken to ensure an impartial physicochemical comparison, as limiting our selection solely to purine metabolites would have introduced bias by disregarding the need to differentiate between molecules with varying characteristics.

### 4.4. Target Prediction According to the Structural Characteristics of Cordycepin and Its Derivatives

The metabolites included in our database underwent target prediction analysis in the SwissTargetPrediction tool (http://www.swisstargetprediction.ch/ (accessed on 15 April 2023) [61]. This tool predicts the potential target proteins of a molecule based on its 2D and 3D similarity to known ligands of those targets. SwissTargetPrediction has an algorithm with a database of 370,000 molecules with known activity and more than 3000 macromolecules from three different species that can be chosen for analysis. The probability score is one of the two parameters obtained in this tool, a combined score of 2D and 3D similar to the input molecule and the algorithms. The second parameter in the output interface is the agreement with the known active compounds (3D/2D). This methodology predicted the most likely targets for each candidate molecule, and the probability score and known targets were collected. The results of the prediction were contrasted with reports from the literature. However, according to the SwissTargetPrediction tool, COR and 3′-dINO do not have similar actives found, and no targets were predicted. To complement this, we employed the Similarity ensemble approach (SEA) tool for COR and 3′-dINO. Like SwissTargetPrediction, this tool predicts potential target proteins based on different types of molecular fingerprints.

### 4.5. Comparative Analysis of the Affinity and Binding Modes of Cordycepin and Its Derivatives with Endogenous Metabolite Targets

For this methodology, the candidate molecules were constructed in Avogadro, according to the SMILES collected, considering a physiological pH of 7.4, and their conformation was optimized using the MMFF94 force field. The resulting conformations were saved as .mol2 files [122]. The protein files used in the simulation were obtained from the Protein Data Bank (Appendix A) in PDB format, filtering the results for *Homo sapiens* [131]. Ligands and other molecules, such as water and ions, were removed from the PDB file using UCSF Chimera 1.16 (University of California, San Francisco, CA, USA) [132]. For affinity comparison, we used Autodock Dock 4.2 to simulate the molecular docking of COR (and its derivatives) in different targets of purine metabolism. In the same way, the interaction of endogenous ligands with these targets was simulated. Autodock Dock Tools 1.5.7 (The Scripps Research Institute, La Jolla, CA 92037, USA) was used as the graphical interface [133]. In the docking study, we added Kollman charges to proteins and Gasteiger charges to ligands. The UniProt database and the DoGSiteScorer tool were used to predict the docking sites of each protein according to its corresponding ligand [134,135]. The docking sites were validated by redocking the ligand from the original PDB file. Proteins, such as ADK and NT5C2 had two different docking sites and are specified in the validation coordinates in the Appendix A. The Lamarckian Genetic Algorithm (GA) was the parameter selected to look for the most stable conformation in the simulation, where the number of GA executions was 10, the maximum number of evaluations was 2,500,000, the maximum number of generations was 270,000, and the gene mutation rate was 0.02. The highest and lowest docking scores obtained by molecular coupling were collected. Furthermore, the lowest docking scores and inhibition constants were collected for each of the metabolites.

Once the molecular docking results were obtained, the binding comparison was carried out. This analysis aimed to observe whether COR and its metabolites had a similar binding mode in the targets evaluated compared to endogenous ligands. First, we generated the .pdb files of those conformations that showed the best affinity. Interactions between endogenous metabolites, COR, and its derivatives at their corresponding targets were observed using Chimera X 1.2.5 (University of California, San Francisco, CA, USA) [136]. 

### 4.6. Molecular Dynamics

#### 4.6.1. Construction of Models by Homology

##### Protein Structure

The canonical sequences are available in the UniProt database and were used for two types of *Homo sapiens* type 3 adenylate cyclase: adenylate cyclase 5 (ADCY05, code O95622) and adenylate cyclase 10 (ADCY10, code Q96PN6). These sequences were subjected to the RosettaFold algorithm [137], available on the Robetta server, and [138] to obtain a three-dimensional model employing molecular homology modeling. In addition, the following modifications were made:ADCY05: Since section 1–195 is identified as a disordered region by UniProt, it is difficult to model and was removed from the prediction in Robetta.ADCY10: Due to the total size of the protein (>1600 aa), it could not be directly modeled, so three segments (1–900 aa, 600–1000 aa, and 700–1610 aa) were modeled in Robetta. Once the models were obtained, they were aligned in the Schrödinger-Maestro 2020-4 program [139] using the 600–1000 aa segment as the alignment standard to build the chimeric model.

##### Structure of Ligands

Based on the nucleotide structure available in crystallographic data, ATP and COR-TP structures were constructed in Schrödinger-Maestro 2020-4. The GTP structure available in PDB 6R4O [140] was used to localize the catalytic site of ADCY05 and subsequently modify the crystallographic nucleotide to obtain ATP and COR-TP. In the case of ADCY10, the ATP available in PDB 8QFF [141] was used to localize the catalytic site and modify the nucleotide to obtain COR-TP.

#### 4.6.2. Molecular Dynamics Simulations

Both the non-ligand ADCY05 and ADCY10 models, as well as the complex model with ATP and COR-TP (6 systems in total), were subjected to MD simulations using Desmond 3,6 [142] available in the Schrödinger-Maestro 2020-4 program (LLC, New York, NY, USA). The force field OPLS2005173 with the polarized 3-point water model (TIP3P) and a final system concentration of 0.15 M adjusted with sodium and chlorine ions were used for all systems. The standard relaxation protocol suggested by Desmond was used, which consists of a series of 6 short simulations with movement restrictions and increasing the temperature until a last production step of 100 ns at 310.15 K and 1 atm of pressure in an isobaric ensemble, surface tension and temperature (NTγP) for ADCY5, and isothermal-isobaric (NTP) for ADCY10. To control these macroscopic variables, the Noose–Hoover thermostat [143] and the Martyna–Tobias–Klein barostat [144] were used, with an integration step of 2 fs and a Coulombic interaction cut-off radius of 9Å. Only for ADCY5 systems, a homogeneous phosphatidylcholine (POPC) membrane was used, focused on the amino acids belonging to the transmembrane α-helix domains according to UniProt.

Once the production simulations were completed, the conformations of the trajectory based on the Root Mean Square Deviation (RMSD) of the position of its α carbons (Cα-RMSD) were grouped up to a maximum of 10 clusters to find the representative structure of the simulation (C1).

#### 4.6.3. Well-Tempered Metadynamics

Using the C1 structures of adenylate cyclase–nucleotide complexes, the systems were prepared to perform metadynamics in Desmond 3.6. The systems were constructed identically to systems for MD simulations and subjected to an identical relaxation protocol. For metadynamics, the distance of the nucleotide’s centers of mass and the amino acids that interact with it was taken as a collective variable, according to UniProt. This collective variable would have a Gaussian repulsion potential amplitude of 0.05 Å and a barrier of 30 Å in a 25 ns production simulation with a Gaussian repulsion potential height of 0.3 kcal/mol and an interval of 0.09 ps in which the Gaussian potential is added during the simulation.

## 5. Conclusions

The biology of cancerous processes is very complex, and cells adapt by different mechanisms to survive and can use alternative signaling pathways to maintain the processes previously treated. However, COR, a product of the complex interaction between organisms, is proposed as a molecule that does not act by a single mechanism nor targets a specific tissue, which confers an advantage in the fight against cancer. In this study, we used chemoinformatics and bioinformatics tools to verify that COR and its derivatives have a strong structural and physicochemical similarity with endogenous metabolites so that they can interact with the same targets. In addition, we tested with MD simulations that COR-TP can stably bind to two isoforms of the enzyme adenylate cyclase, potentially strongly affecting the signaling and plasticity of cells in the tumor microenvironment. Although further experimental evidence is needed, the mechanisms proposed here are helpful in further elucidating the mechanism of action of COR in cancer and other pathologies. 

## Figures and Tables

**Figure 1 ijms-25-05692-f001:**
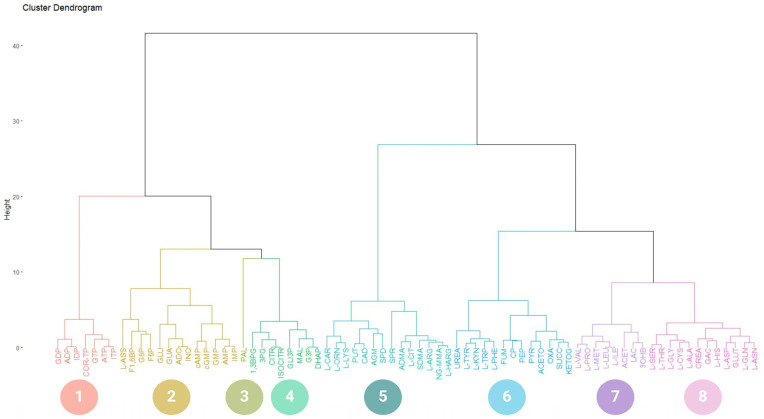
Hierarchical clustering of COR-TP. Molecules clustered according to their physicochemical characteristics. 1. Di- and triphosphate nucleotides. 2. Glucose-related metabolic intermediates, non-phosphorylated purine metabolites, monophosphate nucleotides, arginine-derived metabolites, and cyclic nucleotides. 3. Palmitic acid. 4. Intermediary metabolites of glycolysis and TCA cycle. 5. Polyamines and other arginine-derived metabolites. 6. Aromatic amino acids and other intermediates of the TCA cycle. 7. Branched-chain amino acids and ketone bodies. 8. Other amino acids and creatine-related metabolites. ATP: adenosine triphosphate; ADP: adenosine diphosphate; AMP: adenosine monophosphate; cAMP: cyclic adenosine monophosphate; ADO: adenosine; COR-TP: cordycepin triphosphate; INO: inosine; ITP: inosine triphosphate; IDP: inosine diphosphate; IMP: inosine monophosphate; GUA: guanosine; GTP: guanosine triphosphate; GDP: guanosine diphosphate; GMP: guanosine monophosphate, cGMP: cyclic guanosine monophosphate; ADMA: asymmetric dimethylarginine; SDMA: symmetric dimethylarginine; L-ASS: L-argininesuccinate; F1,6BP: fructose-1,6-bisphosphate; G6P: glucose-6-phospate; F6P: fructose-6-phospate; GLU: glucose; FUM: fumarate; CP: creatine phosphate; PEP: phosphoenolpyruvate; PYR: pyruvate; ACETO: acetoacetate; OXA: oxaloacetate; SUCC: succinate; KETOG: alpha-ketoglutarate; PAL: palmitic acid; NG-MMA: NG-monomethyl-L-arginine; L-HARG: L-homoarginine; 1,3BPG: 1,3-bisphosphoglycerate; 3PG: 3-phosphoglycerate; CITR: citrate; ISOCITR: isocitrate; GLI3P: glycerol-3-phosphate; MAL: malate; G3P: glyceraldehyde 3-phosphate; DHAP: dihydroxyacetone; L-CAR: L-carnitine; L-ORN: L-ornithine; L-LYS: L-lysine; PUT: putrescine; CAD: cadaverine; AGM: agmatine; SPD: spermidine; SPR: spermine; L-CIT: L-citrulline; L-ARG: arginine; L-ALA: L-alanine; L-ASN: L-asparagine; L-ASP: L-aspartate; L-CYS: L-cysteine; L-GLN: L-glutamine; GLUT: L-glutamate; L-GLY: L-glycine; L-HIS: L-histidine; L-ILE: L-isoleucine; L-LEU: L-leucine; L-MET: L-methionine; L-PHE: L-phenylalanine; L-PRO: L-proline; L-SER: L-serine; L-THR: L-threonine; L-TRP: L-tryptophan; L-TYR: L-tyrosine; L-KYN: L-kynurenine; L-VAL: L-valine, CREA: creatine; GAC: guanidinoacetate; LAC: lactate; ACET: acetate; 3OHB: 3-O-hydroxybutyrate.

**Figure 2 ijms-25-05692-f002:**
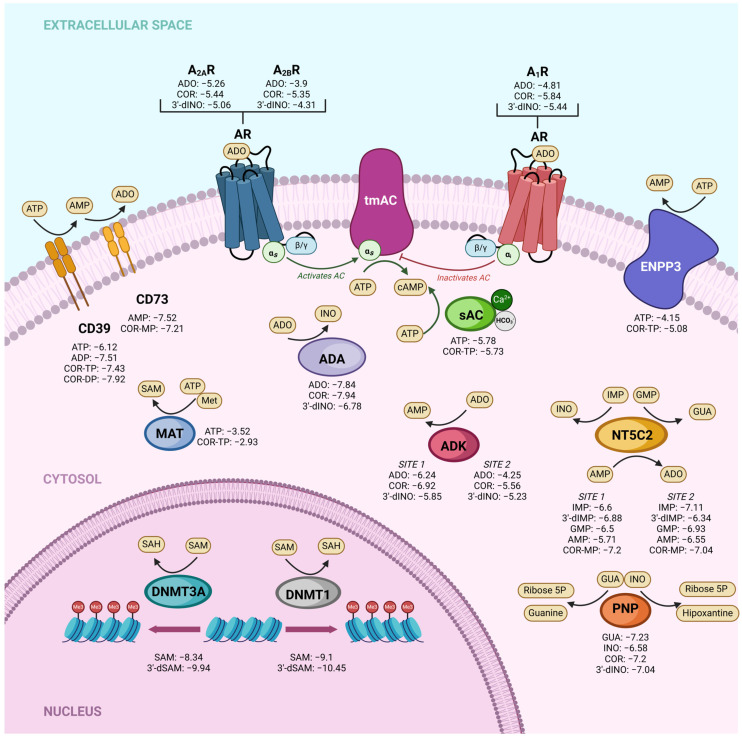
Docking scores (kcal/mol) of different targets with COR and derivatives compared to affinities of endogenous metabolites. ADK: adenosine kinase; sAC: soluble adenylate cyclase; tmAC: transmembrane adenylate cyclase; PNP: purine nucleoside phosphorylase; MAT: methionine adenosyl transferase; ADA: adenosine deaminase; NT5C2: cytosolic 5′-nucleotidase II; ENPP3: ectonucleotide pyrophosphatase/phosphodiesterase 3; DNMT3A: DNA methyltransferase 3 alpha; DNMT1: DNA methyltransferase 1; A_1_R: adenosine receptor 1; A_2A_R: Adenosine receptor 2A; A_2B_R: Adenosine receptor 2B.

**Figure 3 ijms-25-05692-f003:**
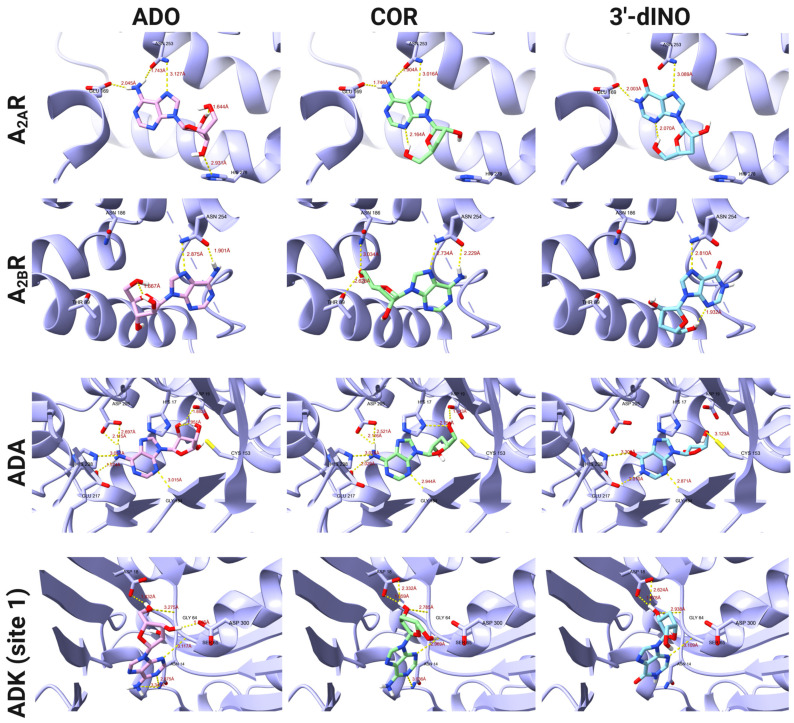
Binding modes of COR, 3′-dINO, and ADO with different targets like ADK (site 1), ADA, A_2A_R, and A_2B_R.

**Figure 4 ijms-25-05692-f004:**
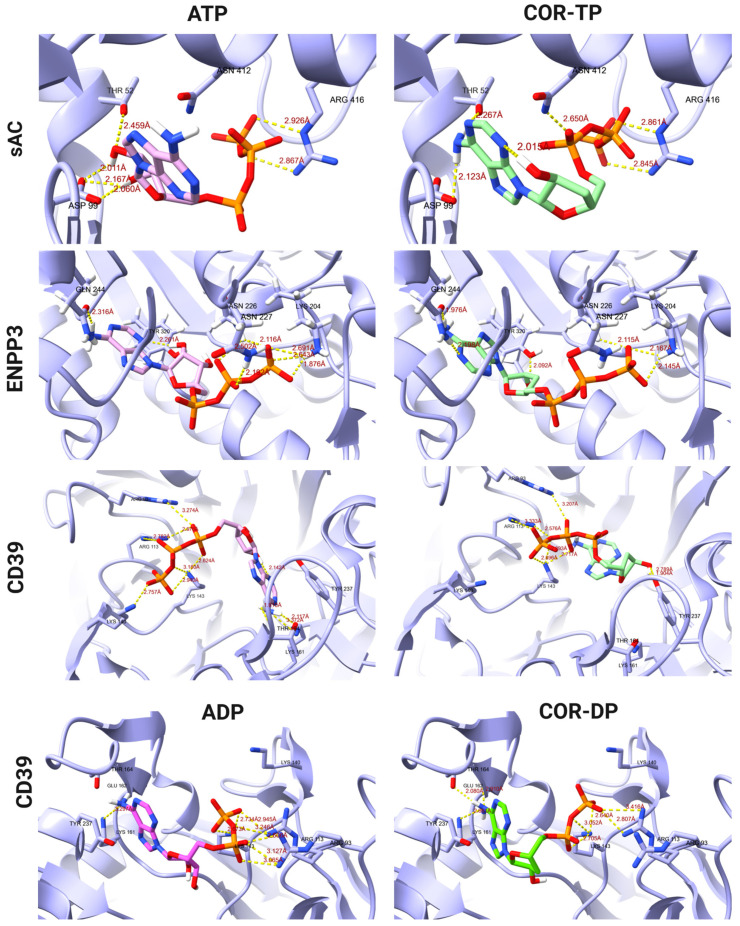
Binding modes of ATP and COR-TP with different targets like CD39, ENPP3, and sAC, as well as ADP, and COR-DP with CD39.

**Figure 5 ijms-25-05692-f005:**
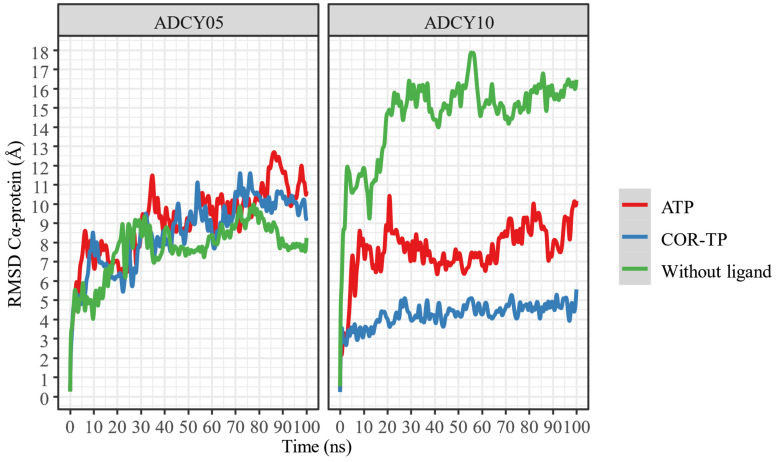
RMSD of the α carbons of the protein–ligand complexes, compared to protein alone.

**Figure 6 ijms-25-05692-f006:**
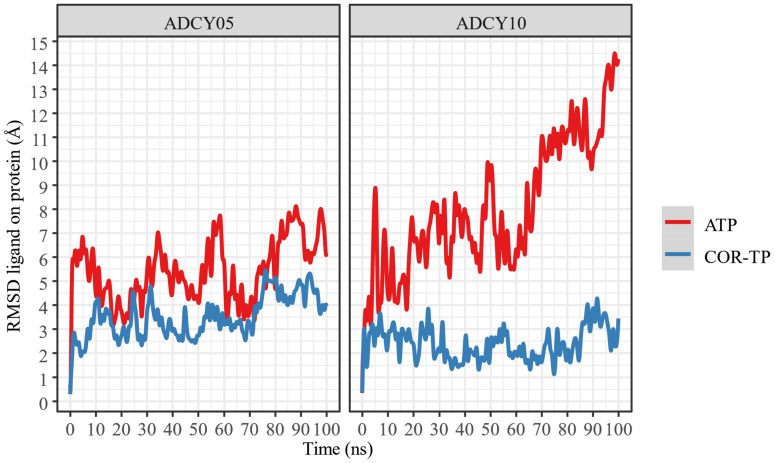
RMSD of ligand atoms relative to protein.

**Figure 7 ijms-25-05692-f007:**
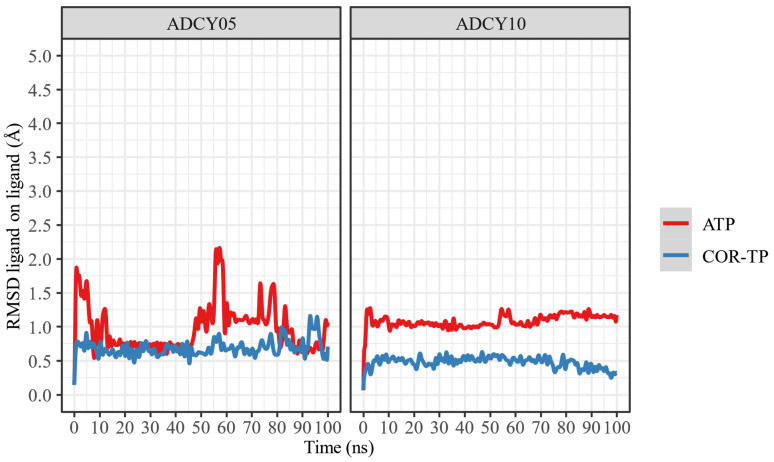
RMSD of the ligand atoms relative to their original position.

**Figure 8 ijms-25-05692-f008:**
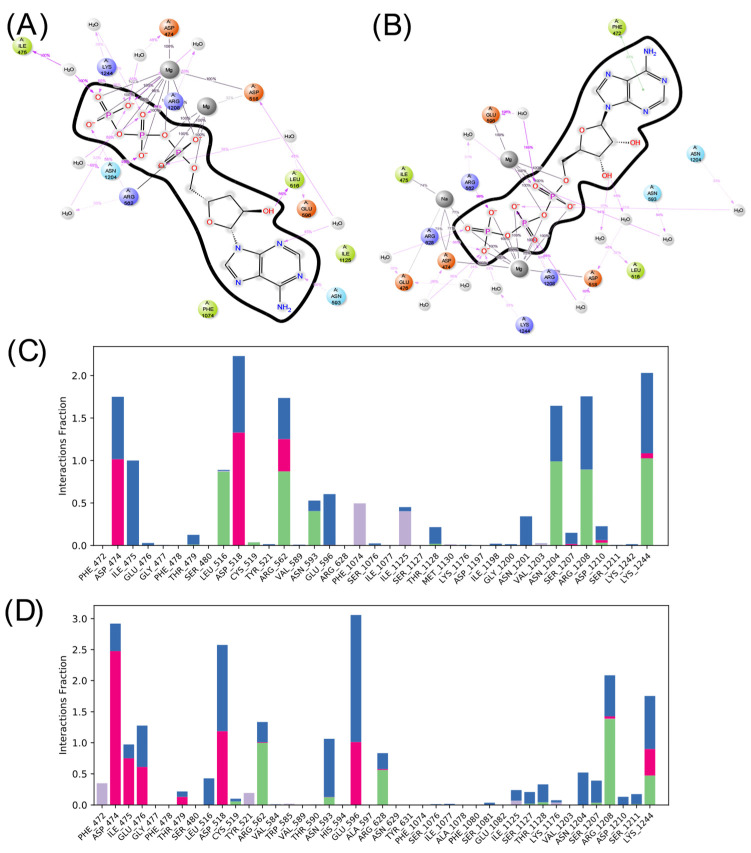
Summary of ligand interactions with ADCY05. (**A**) COR-TP interactions; (**B**) ATP interactions. (**C**) Amino acid residues with which COR-TP interacts; (**D**) amino acid residues with which ATP interacts. Green represents hydrogen bonds; purple represents hydrophobic interactions; pink represents ionic interactions; and blue represents water-mediated interactions.

**Figure 9 ijms-25-05692-f009:**
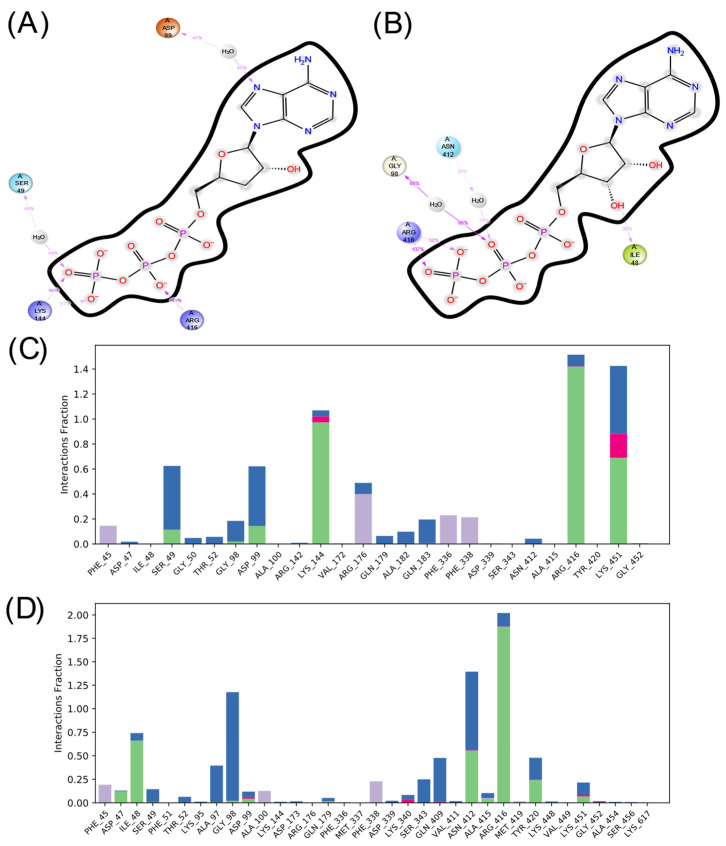
Summary of ligand interactions with ADCY10. (**A**) COR-TP interactions; (**B**) ATP interactions. (**C**) Amino acid residues with which COR-TP interacts; (**D**) amino acid residues with which ATP interacts. Green represents hydrogen bonds; purple represents hydrophobic interactions; pink represents ionic interactions; and blue represents water-mediated interactions.

**Figure 10 ijms-25-05692-f010:**
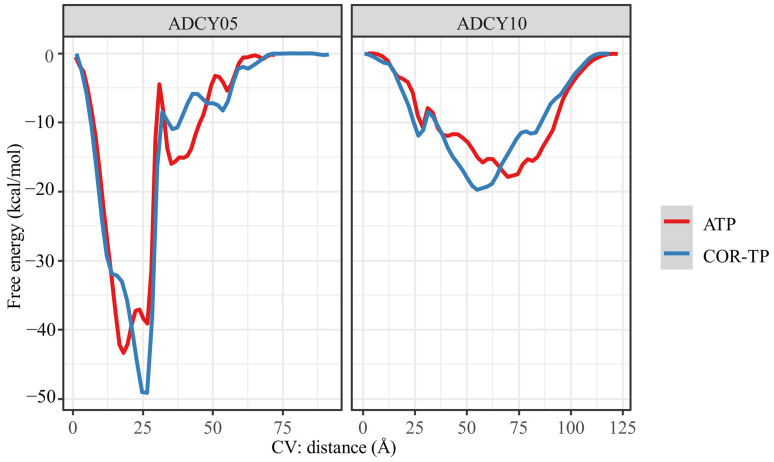
Free energy profile obtained by metadynamics using the distance of the ligand-catalytic site centers of mass as a collective variable.

**Figure 11 ijms-25-05692-f011:**
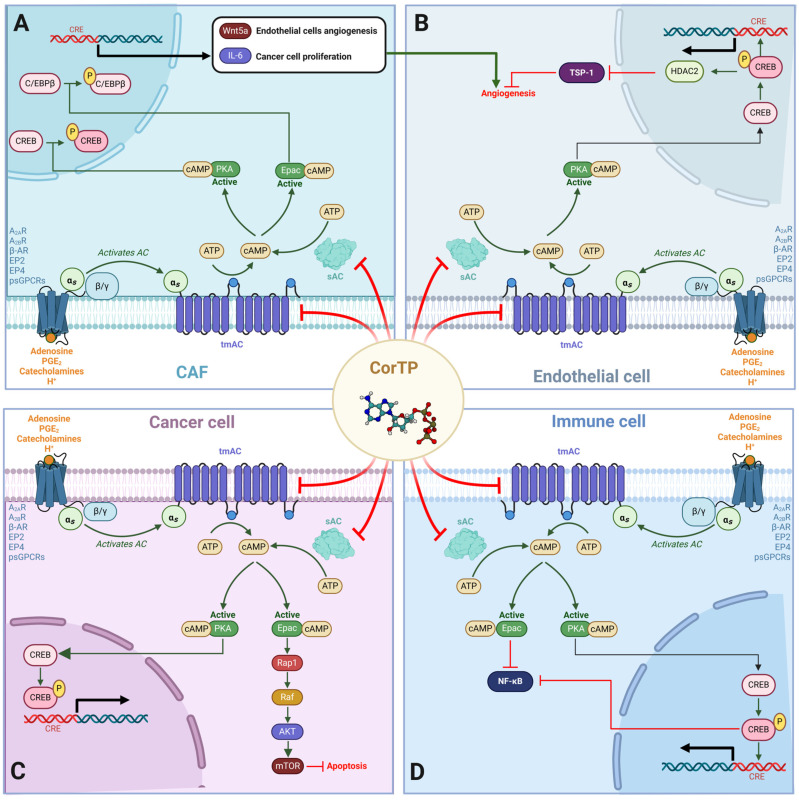
Effects of cAMP-mediated signaling on TME that COR-TP could disrupt by decreasing tmAC and sAC activity. (**A**) In CAF, cAMP-mediated signaling can activate PKA and Epac, which can phosphorylate CREB and C/EBPβ, respectively. This increases the expression of IL-6 and Wnt5a, which favors the proliferation of tumor cells and angiogenesis in endothelial cells. (**B**) In endothelial cells, the cAMP-PKA-CREB axis induces the activation of HDAC2, which represses the expression of TSP-1, a potent angiogenesis inhibitor. (**C**) In tumor cells, cAMP activates PKA and Epac. PKA activates CREB, leading to the transcription of genes involved in proliferation and survival. Epac, in turn, activates Rap1, which activates the MAPK pathway and mTOR through AKT, inhibiting apoptosis. (**D**) In immune system cells, cAMP signaling activates Epac, inhibiting NF-κB. The cAMP-PKA-CREB axis also favors the inhibition of NF-κB, decreasing the proliferation, activation, and release of pro-inflammatory cytokines. In addition, tolerogenic processes in immune cells are favored, significantly reducing their antitumor activity. psGPCRs: proton-sensing G protein-coupled receptors; A_2A_R: Adenosine receptor 2A; A_2B_R: Adenosine receptor 2B; EP2: Prostaglandin E2 (PGE_2_) receptor 2; EP4: Prostaglandin E2 (PGE_2_) receptor 4; β-AR: β-adrenergic receptor; TSP-1: thrombospondin 1; CREB: cAMP response element-binding; C/EBPβ: CCAAT/enhancer-binding protein beta; Wnt5a: Wnt Family Member 5A; CRE: cAMP response element; sAC: soluble adenylate cyclase; tmAC: transmembrane adenylate cyclase.

**Table 1 ijms-25-05692-t001:** Comparison between adenosine and reported cordycepin derivatives. ATP: adenosine triphosphate; ADP: adenosine diphosphate; AMP: adenosine monophosphate; ADO: adenosine; INO: inosine; COR-TP: cordycepin triphosphate; COR-DP: cordycepin diphosphate; COR-MP: cordycepin monophosphate; COR: cordycepin; 3′-dINO: 3′-deoxyinosine; SAM: s-adenosyl methionine; 3′-dSAM: 3′-deoxy-s-adenosyl methionine.

Adenosine Derivatives	Reported Cordycepin Derivatives
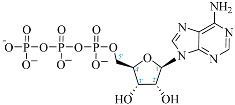 ATP	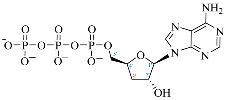 COR-TP
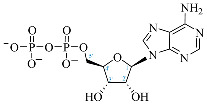 ADP	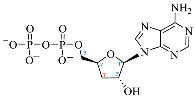 COR-DP
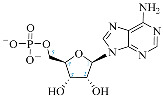 AMP	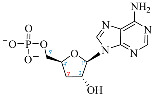 COR-MP
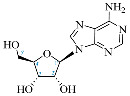 ADO	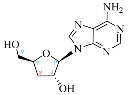 COR
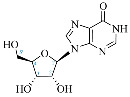 INO	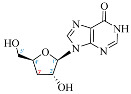 3′-dINO
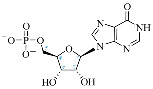 IMP	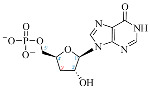 3′-dIMP
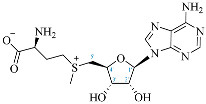 SAM	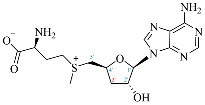 3′-dSAM

**Table 2 ijms-25-05692-t002:** Structural comparison at 2D and 3D level between COR and derivatives against adenosine and derivatives. TC: Tanimoto coefficient; CT: Color Tanimoto; ST: Shape Tanimoto; and ComT: Combo T. High scores of TC, ST, or CT are highlighted in bold.

	COR-TP				COR-DP				COR-MP			
	TC	ST	CT	ComT	TC	ST	CT	ComT	TC	ST	CT	ComT
ATP	**99**	**86**	**43**	**129**	98	86	47	133	97	77	55	132
ADP	98	82	44	126	**99**	**77**	**55**	**132**	97	88	56	144
AMP	97	72	44	116	97	82	47	129	**99**	**91**	**65**	**156**
ADO	93	62	38	100	93	75	51	126	95	87	52	139
INO	73	59	14	73	73	69	23	92	74	87	29	116
	**COR**				**3′-dINO**				**ADO**			
	**TC**	**ST**	**CT**	**ComT**	**TC**	**ST**	**CT**	**ComT**	**TC**	**ST**	**CT**	**ComT**
ATP	92	66	45	111	73	60	19	79	93	67	56	123
ADP	93	75	50	125	73	75	29	104	94	73	60	133
AMP	94	85	58	143	74	82	25	107	95	88	67	155
ADO	**98**	**98**	**85**	**183**	77	98	52	150	**100**	**100**	**100**	**200**
INO	77	98	52	150	**99**	**94**	**63**	**157**	78	89	34	123

**Table 3 ijms-25-05692-t003:** Models obtained by homology modeling and quality control.

System	Adenylate Cyclase Type 05 (tmAC)	Adenylate Cyclase Type 10 (sAC)
Model	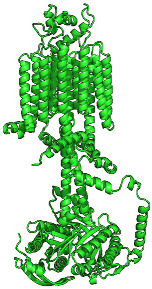	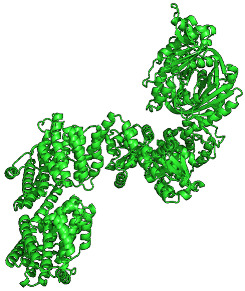
Favored Rotamers	99.24%	99.45%
Ramachandran Plot	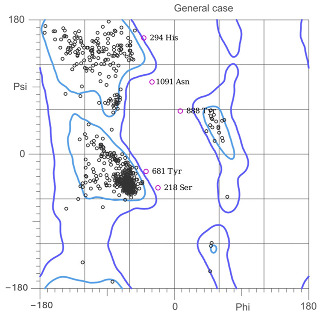	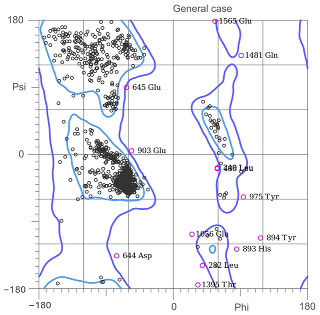
Ramachandran not allowed	0.75%	1.06%
Ramachandran, favored	96.23%	96.39%
Z-score	0.95 ± 0.25	0.14 ± 0.20

**Table 4 ijms-25-05692-t004:** Clustering results per trajectory.

Protein	Ligand	Clusters	Histogram Frequency	C1 Timeframe (ns)
ADCY5	COR-TP	21	21, 16, 14, 14, 12, 12, 11, 10, 10, 10, 9, 9, 8, 8, 7, 6, 6, 5, 5, 4, 3	92.5
	ATP	22	15, 14, 14, 13, 13, 12, 12, 11, 11, 11, 9, 9, 9, 9, 8, 6, 6, 5, 4, 3, 3, 3	84.0
ADCY10	COR-TP	21	18, 14, 12, 12, 11, 11, 11, 10, 10, 10, 10, 10, 9, 8, 8, 8, 7, 7, 6, 4, 4	72.0
	ATP	21	19, 19, 18, 16, 12, 12, 11, 11, 9, 9, 8, 8, 7, 7, 7, 7, 6, 4, 4, 4, 2	52.0

**Table 5 ijms-25-05692-t005:** Free binding energy (kcal/mol) of ligand–adenylate cyclase complexes.

	Ligand	COR-TP	ATP
Protein			
ADCY05		−51.92	−44.39
ADCY10		−20.38	−19.41

## Data Availability

Data is contained within the article and Appendix A.

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
