# Peer review of "Cordycepin Triphosphate as a Potential Modulator of Cellular Plasticity in Cancer via cAMP-Dependent Pathways: An In Silico Approach"

_ijms, 2024, doi:10.3390/ijms25115692_

Round 1

Reviewer 1 Report

Comments and Suggestions for Authors

In this study, the authors have employed various computational methodologies to elucidate the molecular mechanisms of cordycepin and its derivatives. The rationale behind their approach appears sound, and their workflow is comprehensive. They offer thorough explanations and support their findings with relevant literature, although the lack of biological evaluation of their hypothesis is notable. However, certain crucial aspects have been overlooked by the authors, which could potentially impact the validity of their results and lead to misleading conclusions. The reviewer recommends a thorough re-evaluation of their computational work, particularly in the area of molecular docking. At this stage, the manuscript requires significant revisions. Specific comments are provided below:

-- It is widely recognized that ions play a crucial role in the binding of molecules such as ATP and ADP. The authors should justify the absence of ions in their molecular docking protocol. This omission poses a significant concern and could greatly impact the outcomes, potentially leading to misleading conclusions.

-- Figure 5: The protein-protein/ligand complexes appear to be unstable, as evidenced by the high RMSD values (>4 Å), indicating system instability. This issue may stem from selecting an incorrect docked pose prior to initiating the MD simulations. The authors should investigate this further.

-- Authors should move Table 4 containing the chemical structures to the database creation (Section 2.1) for better readability.

-- In the introduction, lines 122-127: it is not clear why the specific example of identification of COX-2 inhibitors is cited to provide the significance of employing bioinformatic and chemoinformatic approaches. There are plenty of examples in literature and these should be cited together or the authors should cite studies related to them that have employed these approaches.

-- Line 182-183: According to the literature, ST, CT, and combo T values above 80, 50, and 130 suggest acceptable similarities – no references cited.

-- Add figure/table numbers when discussing them.

-- Figures 3, 4, and 9 show the ligand-protein interactions: label the amino acid residues. It is impossible to read them.

-- The last sentence in the conclusions is irrelevant to the current study: Therefore, we suggest the continuation of the clinical use of cordycepin as an optimal sensitizer and adjuvant on current therapies with which it has already been shown to be useful, such as immunotherapy, and as a molecule with a synergistic effect with beta-blockers and other small molecules that affect the metabolism of cancer cells.

-- The authors use terms like ‘molecular docking assay’ and ‘molecular dynamics assay’ throughout the manuscript. These should be corrected to molecular docking 'studies' and molecular dynamics ‘simulations’.

Author Response

Dear Reviewer,

I hope this message finds you well. I would like to express my sincere gratitude for the time and effort you have dedicated to reviewing our manuscript. Your insightful comments and suggestions have greatly contributed to improving the quality of our work. We have carefully considered each point and have made revisions accordingly.

Thank you once again for your valuable contribution to our research.

Next, I would like to list your comments and our response or actions implemented in the manuscript:

1) It is widely recognized that ions play a crucial role in the binding of molecules such as ATP and ADP. The authors should justify the absence of ions in their molecular docking protocol. This omission poses a significant concern and could greatly impact the outcomes, potentially leading to misleading conclusions.

Answer: Regarding our molecular docking studies performed without magnesium ions, it's important to mention that our focus was not on determining the exact binding mode of these ligands. It is important to reiterate that the hypothesis driving our study was predicated on the high structural and physiochemical similarities between cordycepin (and its derivatives) and the purine-derived endogenous metabolites, which we postulated could be extrapolated to comparable affinities and binding modes in their respective targets. Consequently, our primary objective was to ascertain whether the binding modes between the di- and triphosphorylated nucleotides (ADP or ATP) were analogous to the corresponding versions of cordycepin (COR-DP or COR-TP). Therefore, we did not include magnesium ions in these simulations. Despite these limitations in our methodology, we opted to conduct molecular dynamics simulations of the adenylate cyclases with ATP or COR-TP, including the magnesium ions, in an effort to account for the influence of this metal on binding modes. Moreover, our simulations were not confined to a single target but were conducted on both the cytosolic and transmembrane isoforms, yielding consistent results across both proteins, even in the presence of magnesium. In response to the reviewers’ recommendations, we have chosen to incorporate these limitations of the molecular docking studies into the discussion section to provide a clearer understanding of our results. Currently, our research group has a limitation in conducting numerous molecular dynamics simulations for all required targets (primarily those working with nucleotides of more than one phosphate group) due to computational constraints. Additionally, Autodock 4.2 does not support simulation with metals, and in some instances, the crystallized proteins lacked the original ions.

2) Figure 5: The protein-protein/ligand complexes appear to be unstable, as evidenced by the high RMSD values (>4 Å), indicating system instability. This issue may stem from selecting an incorrect docked pose prior to initiating the MD simulations. The authors should investigate this further.

Answer: 

: While it is accurate to classify a Root Mean Square Deviation (RMSD) greater than 4 Å as indicative of an unstable simulation, this metric reflects the positional shift of atoms relative to a reference structure, in this instance, the initial structure. As delineated in the original manuscript, deviations from this initial structure are present; however, the RMSD trajectory of alpha carbons does not consistently ascend or exceed 4 Å for all simulations after 50-60 ns, thereby indicating structural stabilization after this time of the simulation. Furthermore, the manuscript specifies the execution of metadynamics simulations to compute the free energy profile derived from the representative structure of the simulation. This methodology navigates the potential energy landscape via the modulation of a variable, such as the distance between centers of mass. The time of the simulation that would yield the representative structure for each scenario is delineated below:

    ADCY5-COR_TP:              92.5 ns
    ADCY5-ATP:                     84.0 ns
    ADCY10-COR_TP:            72.0 ns
    ADCY10-ATP:                   52.0 ns

In every instance, the representative structures (C1) of the simulations reside within the stable region (exhibiting no alterations exceeding 4 Å) of the trajectory for each complex. Details about the extraction of the representative structure from the simulation have been incorporated into the manuscript for enhanced clarity. Conversely, concerning the pose employed for these simulations, a review of Figure 7 reveals that the RMSD of the ligands, relative to the initial pose, does not surpass 2 Å. Therefore, in terms of RMSD, the ligands maintain stability in the initial pose throughout the 100 ns trajectory.

3) Authors should move Table 4 containing the chemical structures to the database creation (Section 2.1) for better readability.

Answer: The table was moved to the section 2.1.

4)  In the introduction, lines 122-127: it is not clear why the specific example of identification of COX-2 inhibitors is cited to provide the significance of employing bioinformatic and chemoinformatic approaches. There are plenty of examples in literature and these should be cited together or the authors should cite studies related to them that have employed these approaches.

Answer: This part of the text was removed. Instead, we mentioned general examples and the information on COX-2 is now available only as a recommended reference.

4)  Line 182-183: According to the literature, ST, CT, and combo T values above 80, 50, and 130 suggest acceptable similarities – no references cited.

Answer: the reference was added in the specified section

5) Add figure/table numbers when discussing them.

Answer: All table/figure numbers were added according to this indication

6)  Figures 3, 4, and 9 show the ligand-protein interactions: label the amino acid residues. It is impossible to read them.

Answer: The side chains of the amino acids were colored by heteroatom in figures 3 and 4,  and the resolution of the figures was increased. In the case of figures 8 and 9, they were modified for better readability.

7) The last sentence in the conclusions is irrelevant to the current study: Therefore, we suggest the continuation of the clinical use of cordycepin as an optimal sensitizer and adjuvant on current therapies with which it has already been shown to be useful, such as immunotherapy, and as a molecule with a synergistic effect with beta-blockers and other small molecules that affect the metabolism of cancer cells.

Answer: This part was removed from the manuscript.

8) The authors use terms like ‘molecular docking assay’ and ‘molecular dynamics assay’ throughout the manuscript. These should be corrected to molecular docking 'studies' and molecular dynamics ‘simulations’

Answer: The terms were corrected across the whole manuscript

Reviewer 2 Report

Comments and Suggestions for Authors

In this study by Gonzalez-Llerena et. al. the authors investigate potential of Cordycepin (COR), Cordycepin triphosphate and its derivatives as analogues towards modulation of cAMP dependent pathway proteins (especially ADCY05 and ADCY10) (critical in cancers) as the COR only lacks 3’OH in its structure compared to Adenosine. Authors used in sillico approaches like docking, target prediction and Molecular dynamics to access their hypotheses.  Overall, the manuscript should be reproducible using publicly available resources. While COH similarity is not a new idea, but I like this investigation as it sheds light onto a possibility about cellular plasticity in cancer. I feel these results should be tested in vivo and in vitro (not in this manuscript) before it can be accepted by broader scientific community. I have few major and few minor concerns.

 Where the MD simulations run as replicates or single run. Single trajectory analysis is susceptible to overinterpretation and/or under sampling. Authors should show that the extending of simulations and/or running replicates would not benefit the study. Only RMSD analysis on single MD trajectory does not suffice convergence criteria. For example: further increase in MD length (maybe 20-25ns)/running replicates does not lead to new cluster(s) of structures (conformations).

Authors performed docking study with a lot of different combinations. The docking results seem to support author’s hypotheses but the docking methodologies themselves have large error margins. Sometime back I came across (https://doi.org/10.1038/s41597-022-01631-9), in this study, I would like to draw attention to figure 2 where the experimental and predicted affinity are within +-2 kcal/mol for 2000 known binders in RCSB PDB (the study used vina (a variant of Autodock) and MM-PBSA). Therefore, docking should be performed using multiple tools and results preferably interpreted using consensus. At least results from few other docking tools should be included in supplementary section for the benefit of other researchers who can then assess.

[Korlepara, D.B., Vasavi, C.S., Jeurkar, S. et al. PLAS-5k: Dataset of Protein-Ligand Affinities from Molecular Dynamics for Machine Learning Applications. Sci Data 9, 548 (2022).]

Authors performed clustering but did not mention which clustering algorithm was utilized and how many clusters were made (they mention “.. upto a maximum of 10 clusters ..” [line 805]). Additionally, what was the percentage distribution in each cluster.

Figure 5,6 and 7: Authors mention 3dATP,ATP,COR-TP and w/ ligand in a confusing manner. I could not find reference to 3dATP anywhere in the written part of the manuscript. I think they mean 3-deoxy ATP. The section 2.6.2 along with associated figures should be made clear.

Authors used different colors for ligands (sticks) but single color for protein residues. It will be useful if the residues (important residues in stick representation) are colored by atom name (example: oxygen red, nitrogen blue) this will make differentiating between polar and non-polar residues easier thereby making figures reader friendly.

Font size could be increased for critical residues in figures. Example: ligand interaction diagram in Figure 9.

Line 265-266 in not in English.

I appreciate that authors on having done a through literature review. At times it feels like it is a literature review manuscript especially in discussion section. The manuscript has 173 references. This feels overdone, only parts that are directly relevant to results and hypotheses under investigation should be kept in main manuscript. Some parts could be tabulated and put in supplementary section.

Figure 6 and 7: Figure 6 uses red for atp and blue for COR-TP, Figure 7 atp uses blue and COR-TP red. This could be made consistent. Word “System” could be dropped in the figure (a minor suggestion). Sub figures could be marked a and b (in both 6 and 7) and used appropriately in the manuscript writeup to point readers to right figure section.  

Line 388-390 is not clear.

Line 161: pH 7,4 should be pH7.4

Author Response

Dear Reviewer,

I hope this message finds you well. I would like to express my sincere gratitude for the time and effort you have dedicated to reviewing our manuscript. Your insightful comments and suggestions have greatly contributed to improving the quality of our work. We have carefully considered each point and have made revisions accordingly.

Thank you once again for your valuable contribution to our research.

Next, I would like to list your comments and our response or actions implemented in the manuscript:

1) Where the MD simulations run as replicates or single run. Single trajectory analysis is susceptible to overinterpretation and/or under sampling. Authors should show that the extending of simulations and/or running replicates would not benefit the study. Only RMSD analysis on single MD trajectory does not suffice convergence criteria. For example: further increase in MD length (maybe 20-25ns)/running replicates does not lead to new cluster(s) of structures (conformations).

Answer: The authors agreed with the observation regarding the potential overinterpretation and/or undersampling with only one molecular dynamics trajectory of 100 ns. Nevertheless, upon examination of the RMSD trends in Figure 5, it becomes evident that the RMSD of the simulations did not exhibit a fluctuation exceeding 4 Å and entered a phase of stabilization. Therefore, prolonging these simulations would not yield additional information in this context. It is plausible that the execution of replicas of these simulations would enable a more comprehensive understanding of the conformational space of the protein-ligand complexes. However, the primary objective of conducting these molecular dynamics simulations was to identify a representative structure (C1) for the execution of metadynamics simulations, with the ultimate goal of calculating the free energy profile from it. This technique explored the potential energy surface through the alteration of a variable such as the distance between centers of mass. Consequently, the simulation of metadynamics not only facilitates the acquisition of binding energy, which was the final objective of this study to ascertain the viability of COR-TP as a competitive inhibitor of these proteins but also enables a more thorough exploration of the conformational profile of protein-ligand complexes. The text has been revised to enhance clarity in this section.

2) Authors performed docking study with a lot of different combinations. The docking results seem to support author’s hypotheses but the docking methodologies themselves have large error margins. Sometime back I came across (https://doi.org/10.1038/s41597-022-01631-9), in this study, I would like to draw attention to figure 2 where the experimental and predicted affinity are within +-2 kcal/mol for 2000 known binders in RCSB PDB (the study used vina (a variant of Autodock) and MM-PBSA). Therefore, docking should be performed using multiple tools and results preferably interpreted using consensus. At least results from few other docking tools should be included in supplementary section for the benefit of other researchers who can then assess.

[Korlepara, D.B., Vasavi, C.S., Jeurkar, S. et al. PLAS-5k: Dataset of Protein-Ligand Affinities from Molecular Dynamics for Machine Learning Applications. Sci Data 9, 548 (2022).]

Answer: In response to the reviewers’ observations regarding potential variations in predicted affinity values and binding modes when utilizing a single docking tool, we chose to conduct additional docking studies using two alternative software tools, namely AutoDock Vina and SwissDock. Upon completion of these additional dockings, we noted a consistency in affinity values across the different tools. Furthermore, the Root Mean Square Deviation (RMSD) values suggested a comparable similarity in binding modes for most of the ligands. These findings support the assertion that, despite minor variations, the predicted conformations for the phosphorylated derivatives of cordycepin closely resemble those of purine-derived endogenous ligands. These results have been made publicly accessible in the Supplementary Material Part A.

3) Authors performed clustering but did not mention which clustering algorithm was utilized and how many clusters were made (they mention “.. upto a maximum of 10 clusters ..” [line 805]). Additionally, what was the percentage distribution in each cluster.

Answer: The physicochemical comparison results through HC for COR-TP were better described. This description included the distribution percentages, the number of clusters generated, and the metabolites contained in each group, highlighting their biological role. In addition, the complete information for all COR derivatives different from COR-TP was included in the supplementary material. In the methods section, the method used to construct the clusterings was more clearly specified. Finally, it should be mentioned that the following text “upto a maximum of 10 clusters” refers to the cluster of molecular dynamics and not to the hierarchical clustering of the physicochemical comparison.

4) Figure 5,6 and 7: Authors mention 3dATP,ATP,COR-TP and w/ ligand in a confusing manner. I could not find reference to 3dATP anywhere in the written part of the manuscript. I think they mean 3-deoxy ATP. The section 2.6.2 along with associated figures should be made clear.

Answer: All the mentioned figures were corrected and standardized as indicated.

5) Authors used different colors for ligands (sticks) but single color for protein residues. It will be useful if the residues (important residues in stick representation) are colored by atom name (example: oxygen red, nitrogen blue) this will make differentiating between polar and non-polar residues easier thereby making figures reader friendly.

Answer: the side chain of the amino acid residues were colored in figures by heteroatom to improve the understanding of the binding between the protein and the ligand.

6) Font size could be increased for critical residues in figures. Example: ligand interaction diagram in Figure 9.

Answer: The resolution and font size in figure 9 were incremented for better readability.

7) Line 265-266 in not in English.

Answer: The line was written in english

8) I appreciate that authors on having done a through literature review. At times it feels like it is a literature review manuscript especially in discussion section. The manuscript has 173 references. This feels overdone, only parts that are directly relevant to results and hypotheses under investigation should be kept in main manuscript. Some parts could be tabulated and put in supplementary section.

Answer: Some of the references were tabulated and placed in the supplementary material, leaving the most relevant references to support our hypotheses.

9) Figure 6 and 7: Figure 6 uses red for atp and blue for COR-TP, Figure 7 atp uses blue and COR-TP red. This could be made consistent. Word “System” could be dropped in the figure (a minor suggestion). Sub figures could be marked a and b (in both 6 and 7) and used appropriately in the manuscript writeup to point readers to right figure section.

Answer: The color of ligands in figures 6 and 7 was standardized. The word "System" was removed. The figures 6  and 7 were appropriately writeup in the text once they were standardized.

10) Line 388-390 is not clear.

Answer: This line was improved in the manuscript by adding new data about the molecular dynamics to gain a better understandig of the main idea.

11) Line 161: pH 7,4 should be pH7.4

Answer: the comma was changed for a dot.

Round 2

Reviewer 1 Report

Comments and Suggestions for Authors

The authors have addressed all the concerns and the manuscript is okay to be published.

Reviewer 2 Report

Comments and Suggestions for Authors

The authors have made the required modifications and replied to my comments in a satisfactory manner.